# Association of Single Nucleotide Polymorphisms of Cytokine Genes with Depression, Schizophrenia and Bipolar Disorder

**DOI:** 10.3390/genes14071460

**Published:** 2023-07-17

**Authors:** Ekaterina V. Mikhalitskaya, Natalya M. Vyalova, Evgeny A. Ermakov, Lyudmila A. Levchuk, German G. Simutkin, Nikolay A. Bokhan, Svetlana A. Ivanova

**Affiliations:** 1Mental Health Research Institute, Tomsk National Research Medical Center of the Russian Academy of Sciences, 634014 Tomsk, Russia; natarakitina@yandex.ru (N.M.V.); rla2003@list.ru (L.A.L.); ggsimutkin@gmail.com (G.G.S.); bna909@gmail.com (N.A.B.); 2Institute of Chemical Biology and Fundamental Medicine, Siberian Branch of the Russian Academy of Sciences, 630090 Novosibirsk, Russia; evgeny_ermakov@mail.ru

**Keywords:** cytokines, genetic markers, single nucleotide polymorphisms, SNP, genetic association, depressive disorder, bipolar disorder, schizophrenia

## Abstract

Immune gene variants are known to be associated with the risk of psychiatric disorders, their clinical manifestations, and their response to therapy. This narrative review summarizes the current literature over the past decade on the association of polymorphic variants of cytokine genes with risk, severity, and response to treatment for severe mental disorders such as bipolar disorder, depression, and schizophrenia. A search of literature in databases was carried out using keywords related to depressive disorder, bipolar disorder, schizophrenia, inflammation, and cytokines. Gene lists were extracted from publications to identify common genes and pathways for these mental disorders. Associations between polymorphic variants of the *IL1B*, *IL6*, and *TNFA* genes were the most replicated and relevant in depression. Polymorphic variants of the *IL1B*, *IL6*, *IL6R*, *IL10*, *IL17A*, and *TNFA* genes have been associated with schizophrenia. Bipolar disorder has mainly been associated with polymorphic variants of the *IL1B* gene. Interestingly, the *IL6R* gene polymorphism (rs2228145) was associated with all three diseases. Some cytokine genes have also been associated with clinical presentation and response to pharmacotherapy. There is also evidence that some specific polymorphic variants may affect the expression of cytokine genes. Thus, the data from this review indicate a link between neuroinflammation and severe mental disorders.

## 1. Introduction

Mental disorders are considered multifactorial diseases. They manifest as clinically significant impairments in cognition, emotion regulation, and behavior and can range from mild to severe [1]. Serious mental illnesses or severe mental illnesses limit one or more major life activities, interfere with quality of life, and can lead to disability. Mental disorders involve general neurobiological processes, such as disruption of the nervous tissue and the blood-brain barrier, autoimmune processes, and neurodegenerative processes. Glial and neuronal changes, metabolic changes in cellular processes, and molecular mechanisms lead to impaired expression of neurospecific proteins. The mechanism of immune response is manifested in changes in levels of hormones and neuromodulators, disruption of psychoendocrine processes, and neurotransmitter systems. Studies of the pathophysiology of severe mental disorders have traditionally emphasized dysregulation of the glutamatergic and monoaminergic systems. However, the mechanisms that cause these neurotransmitter abnormalities are still not clear. Accumulating evidence suggests the interaction of neuroinflammation with the serotonergic, dopaminergic, and glutamatergic systems and the pathogenic role of neuroinflammation in mental disorders [2,3,4,5].

Neuroinflammation includes changes in microglia, astrocytes, cytokines, and chemokines in the central nervous system. There are over 300 cytokines, including chemokines, interleukins, interferons, and growth factors, which play an important role in regulating immune and inflammatory responses [6]. Inflammation plays a role in psychiatric disorders such as bipolar disorder (BD) [7], major depressive disorder (MDD) [8], substance use disorder [9], schizophrenia (SCZ) [10,11,12] and post-traumatic stress disorder [13]. Emerging research suggests that genetic vulnerability could be involved in immune activation in psychiatric disorders [14,15].

The results of bidirectional Mendelian randomization analysis of genome-wide association studies (GWAS) indicate the causative relationship between inflammatory regulators and the risk of mental disorders including MDD, SCZ, and BD [16]. However, the most replicated and relevant genetic variants of cytokines include polymorphisms in the *IL1B*, *IL6,* and *IL10* genes [17,18]. Many studies have revealed dysregulation of the concentration of genetically associated cytokines in the blood of patients with MDD [19,20,21], BD [22], and SCZ [23,24]. There is evidence that genetic polymorphisms affect the expression of pro-inflammatory cytokine genes [25]. Dysregulation of cytokine gene expression has been identified in mental disorders [26]. Nevertheless, the results of studies of the associations of cytokine gene variants with severe mental disorders are heterogeneous and require systematization.

This review synthesizes the current literature on polymorphic variants of cytokine genes associated with risk, severity, and response to treatment of mental disorders. The review focused on severe psychiatric disorders such as depression, SCZ, and BD. Severe mental disorders are characterized by a high degree of polygenicity and the involvement of many genes associated with the risk of developing these diseases. The shared genetic basis of immune activation and psychiatric disorders is of significant scientific and clinical interest because it may reveal new links between the immune system and psychiatric disorders. Although there is no clearly responsible single-nucleotide polymorphism (SNP), research proposes that polymorphic variants play an essential role in susceptibility to psychiatric disorders.

The aim of the review is to analyze studies of cytokine genes and establish the genetic overlapping or specificity of cytokine networks in severe mental disorders. Understanding the role of neuroinflammation in the etiology of mental disorders is essential to developing targeted and effective treatments and interventions. Knowledge of these interactions can help identify pathogenic clues and develop new preventive and symptomatic treatments. Advances in this field of science will enable the discovery of early biomarkers of mental disorders that will improve diagnostic and treatment outcomes and, consequently, the quality of life of patients.

## 2. Materials and Methods

### 2.1. Inclusion and Exclusion Criteria

The study includes the current literature on the association of polymorphic variants of cytokine genes with risk, severity, and response to treatment for bipolar disorder, depression, and schizophrenia. The review considered original case-control and prospective studies, GWAS, metaanalyses, Mendelian randomization studies on depressive disorders, SCZ, and BD. In addition, we have included studies of comorbid mental disorders with medical illnesses, such as cardiovascular diseases or cancer. We also included pharmacogenetic studies related to neuroinflammation and psychiatric disorders. Most studies analyzed more than one SNP, so they have been repeatedly cited in the text.

We excluded studies that focused on non-humans (rat or mouse models), as well as other types of diseases not comorbid with depression, SCZ, or BD, such as anxiety, neurosis, or neurological disorders.

### 2.2. Search Strategy

This review comprises the studies published in the last 10 years (between January 2013 and March 2023). We limited our review to these years to best demonstrate current data on the genetic associations between neuroinflammation and mental disorders.

For the literature review, we use sources from PubMed, Scopus, and the Web of Science. We focus on the association between genetic polymorphisms of the most reliable cytokines and severe psychiatric disorders, their severity, and their response to treatment. We selected articles that described studies of single nucleotide polymorphisms of cytokine genes in depression, SCZ, and BD. Keywords included the following: “gene” or ‘‘SNPs’’ or ‘‘single nucleotide polymorphisms’’; ‘‘depression’’ or ‘‘depressive disorder’’ or “schizophrenia” or “bipolar disorder”; ‘‘inflammation’’ or ‘‘cytokine’’ or ‘‘interleukin’’.

### 2.3. Assessment of Studies

The titles and abstracts of articles from each search query were examined to identify those that showed the associations of single nucleotide polymorphisms with depressive disorder, SCZ, or BD. After analyzing the full texts of the selected articles, a decision was made to include or exclude the articles from the review.

For the synthesis we tried to group data by pathology at first and then by particular cytokines. A total of 44 articles were selected for this review. However, the resulting bibliography contains more sources to discuss the identified associations and expand the context. From the included studies we collected the following data: study identifications, number of participants and prime study conclusion.

A Venn diagram was used to summarize data on the association of SNPs of cytokine genes with depression, SCZ and BD. The online tool (https://bioinformatics.psb.ugent.be/webtools/Venn/, accessed on 30 May 2023) was used to calculate the intersections of the list of SNPs. Inkscape 1.2.2 (Free Software Foundation, Inc., Boston, MA, USA) has been applied to the graphic design of a Venn diagram.

## 3. Results

The genes encoding cytokines are highly polymorphic. Previously, it was described that single nucleotide polymorphisms can be associated with increased or decreased production of cytokines. Polymorphisms in the promoter region of cytokine genes may result in inter-individual variation in transcription and expression of genes [25,27], thereby affecting the pathogenesis of mental disorders, the prognosis of their course, and the response to pharmacotherapy. Next, we consider the available data on associations of cytokine gene variants with mental disorders in more detail.

### 3.1. Depressive Disorders

Depressive disorders are a heterogeneous group of diseases. The etiology, precise pathophysiological mechanisms, response to treatment, and outcome of depressive disorders are still poorly understood. The cytokine hypothesis of depression proposes that pro-inflammatory cytokines acting as neuromodulators are a key factor mediating behavioral, neuroendocrine, and neurochemical changes in this disease [28].

Identifying a common genetic substrate for depression and immune activation will help unravel the link between neuroinflammation and depression. This section describes the association of inflammatory cytokine genes with depression, and polymorphisms that increase or decrease this association. Current research indicates that individual polymorphisms play a role in susceptibility to depression and outcome.

Gene polymorphisms encoding cytokines and their receptors can influence their functional activity. Available data on the genetic association of SNPs in cytokines and their receptor genes with depression is presented in Table 1.

The *IL1B* gene is the most extensively examined in the field of psychiatry among the cytokine gene polymorphisms. Genetic polymorphisms in the *IL1B* gene have been well studied in depression. IL-1β has been implicated in the pathophysiology of major depression. Recent studies have identified the involvement of the *IL1B* gene in depression [29,34]. Among polymorphisms in the *IL1B* gene, more attention is given to the −511C/T (rs16944). The −511C allele in *IL1B* was associated with higher *ILB* expression [30] and higher IL-1β levels. The −511C/T polymorphism was associated with a depression [31,34]. The risk variant for depression was the CC genotype (*p* = 0.001, OR = 1.9 CI 1.3–2.7) [31]. There is much research dedicated to studying the contribution of *IL1B*-511C/T in the pathogenesis of depression disorders. Thus, it was shown that −511C/T polymorphism is associated with primary depression and post-stroke depression (PSD) at 2 weeks [29], with depression trajectory after acute coronary syndrome [34]. The −511C allele was also associated with more severe depression following chronic interpersonal stress exposure [30], and the T allele was associated with a positive history of major depression [55]. It was shown that depression in acute coronary syndrome was significantly associated with the level of IL-1β and −511T allele [34]. Another study reported that the −511T/T genotype was associated with both depression one week after surgery for breast cancer and persistent depression at one year of follow-up [38].

In a study by McQuaid et al. (2019) [32], it was shown that the severity of depressive symptoms was higher in individuals with the GG genotype of the *IL1B* rs16944 gene polymorphism. These results are consistent with earlier reports that carriers of the GG genotype have a greater severity of depressive symptoms [30]. The rs16944 *IL1B* was associated with childhood abuse as a predictor of depression scores. In particular, after childhood abuse, men carrying the rs16944*GG of *IL1B* showed particularly severe symptoms of depression [32]. In this way, the level of IL-1β and the −511C/T genotype, alone or together, may be biomarkers of depressive disorder. Targeted interventions for people with higher levels of IL-1β and the *IL1B*-511T allele may reduce the risk of depressive disorder [34].

*IL1B* rs1143643 was significantly associated with Mausdley Staging Method scores to determine treatment response in MDD [35]. In another study of depressive patients combined with childhood trauma, rs1143643 did not increase depressive symptoms. However, the minor A allele showed a protective effect against depressive symptoms after recent life stress [33]. Patients with GC and CC genotypes of rs1143623 (*IL1B*-1560G/ C) demonstrated different levels of disease severity as evaluated by the Hamilton Depression Rating Scale [18]. For the +3953C/T polymorphism of *IL1B*, no associations were found with depression in either the acute or chronic phases [34].

The results of the GWAS, published this year, suggest that specific SNPs, including rs2540315 and rs75746675 in the IL-1 receptor gene *IL1R1*, were associated with a rapid (within 240 min) antidepressant effect of ketamine infusion in patients with treatment-resistant depression [37]. An important limitation of the study is the small number of people examined (65 patients with treatment-resistant depression divided into 3 groups).

Interleukin-6 (IL-6) is a potent biomarker for depression, as its elevated plasma levels in patients with clinical depression have been confirmed by a range of studies [19,42,45]. Genetically-predicted IL-6 was associated with major depression in a multivariable mendelian randomization study (OR = 1.08; 95% C.I., 1.03–1.12) [56]. It has been shown that IL-6 levels are correlated with the *IL6*-634C/G polymorphism (rs1800796), and a G to C polymorphism at the −174 position of the *IL6* promoter region appears to affect *IL6* transcription [27,42]. The −572CC genotype and C allele were significantly associated with depression in the Han Chinese population [40].

It is also reported that the −174G/C polymorphism *IL6* in interaction with various stress factors increases the risk of depression and has a greater impact on symptoms measured by the Zung Self-rating Depression Scale [42]. Udina et al. (2013) found that carrying the CC genotype of rs1800795 *IL6* is associated with less severe IFN-α-induced depression and anxiety [44]. Russian researchers report that −174G/C polymorphism was associated with depression comorbid to coronary heart disease. The frequency of the allele G in this group was higher compared with controls [41]. The polymorphic variant rs1800795 (*IL6*-174G/C) is associated with recent stress on current depressive symptoms and is associated with lifetime depression at a nominal significance level [41].

The polymorphic variant rs2228145 (Asp358Ala) of the IL-6 receptor gene (*IL6R*) was associated with a reduced risk of severe depression and/or psychosis; the adjusted 95% odds ratio for patients with the CC genotype compared with the AA genotype was 0.38 (CI 0.15–0.94). This same polymorphic variant was associated with elevated serum IL-6 levels (P = 5.5 × 10^−22^) [47].

Several studies suggest that IL-8 has neuroprotective functions [57]. At the same time, an investigation of *IL8*-251T/A in breast cancer patients found no association between the alleles and depression [38]. Furthermore, no association was found between *IL8*-251T/A and depression in a study of 732 elderly Koreans [58]. In the examination of a symptom in lung cancer, patients with *IL8*-251TT were more likely to experience severe depression but less susceptible to pain or fatigue [59].

Genetic studies of the anti-inflammatory cytokine IL-10 have shown that carrying genotypes GA and GG of rs1554286 *IL10* is a predictor of anxiety (HR 1.85, *p* = 0.019) in early-stage breast cancer patients in China, which could help identify patients at high risk for psychological problems [60]. An investigation of 167 oncology patients showed that the rare AA genotype of rs1518111 was associated with subsyndromal depression [48]. 

A study of 398 breast cancer patients prior to surgery showed that the rs1295686*A of *IL13* was associated with a symptom cluster of pain, fatigue, sleep disturbance, and depression [49].

IL-18 is expressed in the brain, and it is increased in patients with depression [50,61] and influences stress-related susceptibility to mood and anxiety symptoms by changing amygdala reactivity [62]. Reported that polymorphisms *IL18*-607A/C and *IL18*-137C/G were associated with the effects of antidepressant therapy [50]. Thus, patients carrying CA or AA genotypes of −607A/C and patients carrying GC or CC genotypes of −137C/G were significantly more prone to relapse after therapy and presented a significantly lower time to relapse [50]. Recent research on *IL18* polymorphisms (rs187238, rs1946518 and rs1946519) has not found differences between depressive patients and healthy controls [61].

Meta-analysis data showed that the level of tumor necrosis factor-α (TNF-α) increased in MDD [63]. Increased levels of TNF-α may be conditioned by the presence of a range of specific polymorphic variants [64]. Higher TNF-α levels were associated with post-stroke depression at 2 weeks in the presence of the −850T allele [29]. Another study showed that the *TNFA*-857CT genotype was associated with increasing the risk for prenatal depression in a Mexican mestizo population, and the −238GA genotype reduced the risk [51]. Homozygous for the rare allele in rs1799964 *TNFA* belonged to the subsyndromal depressive symptoms in patients with breast cancer [52]. A GWAS showed that among 57 genes and 92 SNPs identified in MDD patients, only rs769178 *TNFA* was related to depression, and it remained significant after correcting for multiple testing [14].

A number of studies have reported that the *TNFA*-308G/A polymorphism (rs1800629) is associated with depression [31] and is a risk factor for suicide attempts in MDD [36]. It was found that the TT genotype and the T allele of rs1799964 (−1031T/C) were associated with low effectiveness of pharmacotherapy, and the CT genotype and C allele were associated with positive responses to the treatment of depressive disorder [18]. The opposite data of the metaanalysis showed that there was no association of the *TNFA*-308G/A alleles or genotypes with poststroke, late-life, maternal, or major depression [65]. Another recent study, including 83 Polish patients, found no statistically significant association between the genotype/allele frequency of *TNFA*-308G/A and *TNFA*-1031T/C and depression [64].

The AG genotype of the rs2166975 *TGFA* was associated with an increased risk of depression development, while the GG genotype of the rs2166975 *TGFA* reduced the risk. That genotype increased the risk of MDD only in the male population [53].

The *TGFB* + 869T/C polymorphism predicts low activity of TGF-β expression. The study in the Bulgarian population revealed a significant prevalence of the TT genotype of the +869T/C polymorphism in patients with depression (41.3%) compared with healthy subjects (21.2%) (*p* = 0.05, OR = 2.62). In addition, the combination of TT-GC genotypes (+869T/C, +915G/C) in the gene is negatively associated with disease recurrence of depression [54]. Genotype AA of rs1800469 *TGFB* was associated with an earlier age of depression onset, while GG genotype increased the severity of the depressive episode [53]. One study showed an association between the rare A allele of rs2229094 *TNFA* and subsyndromal depression [48].

Thus, literature data show that depression is associated with polymorphic variants of cytokines genes. These associations also include the severity of the depressive disorder or the response to therapy. This complements and expands the data on immune dysfunction in this disease. Further research is needed to determine the precise impact of these polymorphisms and to find potential predisposing or protective alleles that can be used as biomarkers for the risk of depression.

### 3.2. Schizophrenia

There are many studies showing a link between changes in the inflammatory system and SCZ [12,66]. However, there are much fewer studies devoted to research on cytokine genes in SCZ (Table 2).

A number of studies demonstrate the association of polymorphic variants of cytokine genes with SCZ. Particularly, the case-control study reported that the A allele and AA genotype of the *IL1A*-889G/A (rs1800587) polymorphism were associated with SCZ in a South Indian population (*p*  =  0.026; OR  =  1.36; CI  =  1.04–1.79) [67]. The analysis of 621 patients with SCZ in the Polish population showed an association between rs4848306 in the *IL1B* gene and SCZ [70]. It was shown that CC and GC genotypes of rs1800796 (*IL6*-572G/C), but not rs1800795 (*IL6*-174G/C), were significantly associated with chronic SCZ in a Han Chinese population. The −572GC genotype may serve as a protective factor for SCZ [40]. On the other side, Srinivas et al. (2016) reported that the G allele of *IL6*-174G/C (rs1800795) was associated with SCZ (*p*  =  0.037). The GG genotype was observed to be in higher frequency in patients in a recessive model (GG vs. GC + CC); this association with SCZ was significant (*p*  =  0.034) [67].

Moreover, there is a trend toward an association of rs1143627, rs16944, and rs1143623 in the *IL1B* gene with the risk of SCZ. Alleles rs1143627*G, rs16944*A, and rs1143623*G were significantly more frequently transmitted by parents to children with SCZ [68].

*IL6*-572G/C (rs1800796) was associated with SCZ at the genotypic level (*p*  =  0.015). In an additive model (GC vs. GG + CC), this association was enhanced further (*p*  =  0.003) [67]. The polymorphisms *IL10*-1082G/A (rs1800896), *IL10*-592C/A (rs1800872), and *IL10*-819T/C (rs1800871) were associated with SCZ, and the ACC haplotype was more prevalent in SCZ in Saudi Arabian patients [74]. One study found a significant association between rs11792633 *IL33* and the risk of SCZ in an Iranian population. CT and TT genotypes significantly decreased the risk of SCZ [78]. The T carriers (CT and TT genotypes) of *TGFB* + 869T/C were significantly more frequent in SCZ (especially in females) than in healthy subjects [72].

Some of the study is dedicated to a paranoid form of SCZ. It was shown that genotype TT and allele T of the *IL2*-330G/T (rs2069762) polymorphism were significantly associated with the paranoid form of SCZ, as well as allele A of rs1800629 *TNFA* polymorphism [69]. A recent study showed that the rs1126647 of *IL8* was a significant risk for SCZ. The TT genotype and T allele at rs1126647 were also associated with a paranoid form of SCZ. Haplotypes TTT, ACT, and TCT at rs4073-rs2227306-rs1126647 in *IL8* were associated with increased risk for paranoid SCZ [57].

A number of researchers reported on association polymorphisms of cytokine genes and clinical characteristics of SCZ. In a study aimed at finding associations between interleukin genes and subdomains of negative symptoms in SCZ, calculated based on the Positive and Negative Syndromes Scale, it was found that the association between the *IL6*-174G/C (rs1800795) polymorphism and AA scores (avolition and apathy) was close to the level of significance. Patients with the *IL6*-174GG genotype had higher scores compared with the AA genotype [73].

A study published in 2022 found that there is a significant main effect of the *IL10*-1082G/A polymorphism (F = 5.56, df = 2, *p* = 0.004) and the *IL10*-592C/A polymorphism (F = 3.48, df = 2, *p* = 0.03) on the AA (avolition and apathy) scores of the Positive and Negative Syndromes Scale in patients with SCZ [73]. Post-hoc analysis (Bonferroni corrected) revealed that the mean score on the AA subdomain was higher in the *IL10*-592AA genotype group compared with the *IL10*-592CC group (*p* = 0.002). Differences between *IL10*-1082G/A genotypes were dose-dependent: the AA score decreased with the number of copies of an A-allele. The lowest score was observed in the group with the *IL10*-1082GG genotype.

The AA genotype of *IL17A*-197G/A (rs2275913) was associated with higher total scores of bizarre behavior and apathy in female SCZ patients [76].

Research of 772 inpatients with SCZ and 775 healthy controls in a Han Chinese population has shown that increased IL-18 serum levels and the *IL18*-607A/C (rs1946518) polymorphism were positively associated with the PANSS (the Positive and Negative Syndrome Scale) general psychopathology subscore and the PANSS total score [77]. Thus, the interaction between increased IL-18 serum levels and the −607A/C polymorphism influenced clinical psychopathological symptoms, and this dependence was present only among patients carrying the C allele.

The polymorphism rs6676671 in *IL10* was associated with the early age of SCZ onset [70].

Transcriptomic coexpression analysis in the human brain revealed that genes most significantly co-expressed with *IL10* were associated with synaptic vesicle transportation. Moreover, both *IL10RA* and *IL10RB* were most significantly co-expressed with genes that regulate inflammation and also with those that participate in synaptic formation [75]. The *IL10*-592C/A genetic variant was more common in SCZ patients than healthy subjects and was associated with lower serum levels of IL-10 and worse attentional performance in these patients. In the healthy human brain, the *IL10* gene and its receptors are involved in the regulation of neuroinflammation and synaptic functions that are important for cognition, and hence their deficiency may contribute to cognitive impairment in SCZ. 

Thus, it has been shown that there are associations between different polymorphic variants of cytokine genes and the SCZ, its forms, and its clinical characteristics, but the data obtained is not always unambiguous. These genetic data link immunoinflammation with the pathogenesis of SCZ. Further research is needed to clarify and expand our understanding of the role of the immune system in the pathogenesis of SCZ.

### 3.3. Bipolar Disorder

There are not many studies of cytokine genes in BD. Available data on the genetic association of SNPs in cytokines and their receptor genes with BD are presented in Table 3.

Most studies in BD are devoted to the *IL1B* gene. Thus, it is shown that the frequency of CC and CT genotypes of *IL1B*-511C/T (rs16944) was significantly different between BD patients and healthy controls (*p* = 0.04 and *p* = 0.02, respectively) [55]. The T allele of *IL1B* + 3954C/T (rs1143634) was significantly more frequent in early-onset BD patients [55]. Pu et al. (2021) also showed association −511C/T with risk of BD [71]. Shonibare et al. (2020) showed a link between *IL1B*-511C/T and brain morphology. The −511C/T was associated with greater lateral occipital cortex surface area and volume in BD adolescents [80]. 

Among the seven polymorphisms of the *IL1B* gene researched by Pu et al. (2021), the minor alleles of five SNPs exhibited significantly increased frequencies in the BD compared with controls (Table 2). Four of the five SNPs (rs16944, *p* = 0.00570, OR = 1.199 for the A allele; rs1143627, *p* = 0.00793, OR = 1.190 for the G allele; rs12621220, *p* = 0.00746, OR = 1.199 for the T allele; rs1143623, *p* = 0.0127, OR = 1.184 for the G allele) were significantly associated with the risk of BD [71].

One study researched the *IL6* and *IL6R* genes. The C allele and CC genotype of *IL6R* rs2228145 were associated with the early onset of BD [81].

Thus, despite the limited literature data, it has been shown that there are associations between polymorphic variants of cytokine genes and the risk of BD. These data are proof of the immunoinflammatory theory of BD pathogenesis. However, given the limited data, new studies on large samples are needed to expand the existing understanding of the role of cytokine gene polymorphisms in BD.

The studied literature sources allowed us to conclude that mental disorders may have common etiological and maintenance processes as well as cognitive-affective, interpersonal, and behavioral features. The general results of the overall sample of studies indicate that three mental illnesses have common genetic associations.

## 4. Discussion

This review synthesizes the current literature on the association between genetic variants of cytokines and mental disorders risk, severity, and response to treatment. This review suggests that common variants of cytokine genes are associated with both immune alterations and the pathogenesis of psychiatric disorders, including depression, BD, and SCZ. It is hypothesized that immunity may cause psychiatric symptoms accompanying various disorders, and inflammation may be associated with dysregulation of the glutamatergic and monoaminergic systems and oxidative damage in psychiatric illness. Thus, immunity can be associated with the pathogenesis of psychiatric symptoms in patients with classical psychiatric disorders. Therefore, the data in this review indicate an association between neuroinflammation and depression, BD, and SCZ [82,83].

All data from this review on the association of SNPs in cytokine genes with depression, SCZ, and BD are summarized in Figure 1. The summed results indicate that three mental illnesses have common genetic associations. In particular, depression and BD share many genetic associations. Interestingly, the *IL6R* gene polymorphism (rs2228145) was associated with all three mental disorders. This may indicate a common immune-related genetic basis for these pathologies.

Accumulated data indicate that mental disorders are accompanied by changes in cytokine levels. For example, the anti-inflammatory cytokine IL-2 is altered in people with schizophrenia [63,84,85]; IL-6 concentrations are associated with depression [42,45,86]; elevated levels of IL-6 are trait markers for BD [87] and schizophrenia [63,88]; elevations of IL-8 in cerebrospinal fluid are seen in patients with schizophrenia [89]; IL-18 is increased in patients with depression [50,61]; the level of TNF-α increased in patients with SCZ, MDD [63] and BD [90]. It is important to note that changes in the expression of cytokine genes may be conditioned by the presence of a range of specific polymorphic variants [64]. Thus, the *IL1B*-511C allele is associated with higher *IL1B* expression [30], the *IL6*-634C/G polymorphism is correlating with IL-6 levels, and the *IL6*-174G/C is affecting its transcription [27,42]; the ACC haplotype (−1082G/A, −592C/A, −819T/C) of *IL10* was associated with intermediate production of IL-10 in SCZ patients [74].

Psychiatric disorders are highly polygenic and likely to be affected by many genetic loci, each of which has little effect [91]. Our study of the literature data from the last 10 years showed that the most replicated and relevant genetic variants of cytokines include polymorphisms in the genes for *IL1B*, *IL6,* and *IL10*. However, as we have found, even for the most replicated findings, there are many inconsistent results in association studies of genetic variants, immune responses, and effects on disease. For example, numerous studies have revealed a relationship between the minor C allele, or CC genotype, of *IL1B*-511C/T and clinical psychopathological symptoms of different forms of depression [29,30,31,34,38]. On the other side, there is another study demonstrating the association of the TT genotype, not the CC, of *IL1B*-511C/T with depression [38].

This diversity of results may be explained, at least in part, by the heterogeneity of the depression immunophenotype, by environmental influences, genes and environment interactions, genetic variants, and gene expression variations [14]. Moreover, the combined effects of gene-environment interactions rather than purely genetic mechanisms play an essential role in neuroinflammatory processes. Thus, as we found, the effects of some SNPs may only become evident in the presence of stressors such as stroke, surgery, or cancer. For example, the *IL6*-174G/C polymorphism in interaction with various stress factors increases the risk of depression [42], and the *IL1B*-511C/T was associated with more severe depression following chronic interpersonal stress exposure [30]. These findings support the putative hypothesis that pro-inflammatory genetic variation increases the risk of stress-induced depression [30].

The range of polymorphic variants of cytokine genes identified in recent years is quite high. An analysis of the literature data leads to the identification of the involvement of some polymorphisms in various pathologies. For example, rs16944 *IL1B* showed a significant association with depression in scientific studies [31,32,34], symptom severity [30], as well as with SCZ [68] and BD [55,71]. The polymorphism rs2228145 in *IL6R* is associated with depression [47] and BD [81]. The *IL6*-174G/C polymorphism was associated with the severity of symptoms of current depressive disorder [30], with recent stress on current depressive symptoms, with lifetime depression [43], and with mean scores on the AA subdomain of the Positive and Negative Syndromes Scale in patients with schizophrenia [73]. It was also found that rs16944 and 1143623 of the *IL1B* gene were associated with both depression [18,29,30,31] and BD [71]. The polymorphic variants of *IL6*, *IL18,* and *TNFA* were associated with depression [39,40,44,50,51] and SCZ [67,69,71,77].

Conversely, there are a number of studies demonstrating the absence of significant associations between polymorphic variants of immunoinflammation genes and different mental disorders [34,38,58,65]. Therefore, these data require replication in other studies.

**Perspectives.** It is known that the effectiveness of treatment depends on the genetic and clinical heterogeneity of patients and the occurrence of side effects [92]. Only about 30 percent of the patients respond to treatment and experience remission. Future pharmacogenetic testing will help determine the person-specific genetic factors that may predict clinical response and side effects [93].

**Limitations of the evidence.** Significant limitations of the mentioned studies are that they had different sample sizes (tens to thousands). Clinical samples in some of the studies were generally small (less than 100 samples) [37,55,78] and were underpowered for reliable detection of associations between neuroinflammation and mental disorders. Future studies with larger sample sizes are needed to explore these associations. Another limitation of these studies is the sample’s clinical heterogeneity from study to study. The next factor in studies heterogeneity is covariates in analyses. Some of the studies do not include factors influencing inflammatation but not related to mental disorders such as age, gender, medication use, alcohol consumption, obesity, race and ethnicity, menopausal status, and other factors. Including these factors in standardization would strengthen research findings and facilitate comparisons of multiple studies and meta-analyses. Moreover, different researchers studied mental disorders, in particular depression, in combination with organic disorders such as cancer or stroke. Therefore, all these factors can probably introduce heterogeneity into the pooled results.

**Limitations of the review.** We focus on the studies published only in the last 10 years. There is much data on the topic published earlier that was not included in the review. We included only three pathologies: depressive disorder, SCZ, and BD. Neither addictive disorders, nor anxiety, nor neurosis, nor other mental disorders or animal models were considered. We included only studies on single nucleotide polymorphisms and did not consider duplications, deletions, and other chromosome mutations.

## 5. Conclusions

Numerous studies demonstrate an association between different severe psychiatric disorders and inflammation-related biomarkers. The shared genetic basis of immune activation and psychiatric disorders may reveal new links between the immune system and psychiatric disorders. This review synthesizes the current literature on the association between polymorphic variants of cytokine genes and mental disorders such as depression, SCZ, and BD, their severity, and their response to treatment. Understanding the effect of minor genetic variants on the immune system and their contribution to the development of psychiatric disorders is important for the identification of vulnerable individuals, the establishment of genetic biomarkers, and the development of new approaches to therapy based on pharmacogenetics.

## Figures and Tables

**Figure 1 genes-14-01460-f001:**
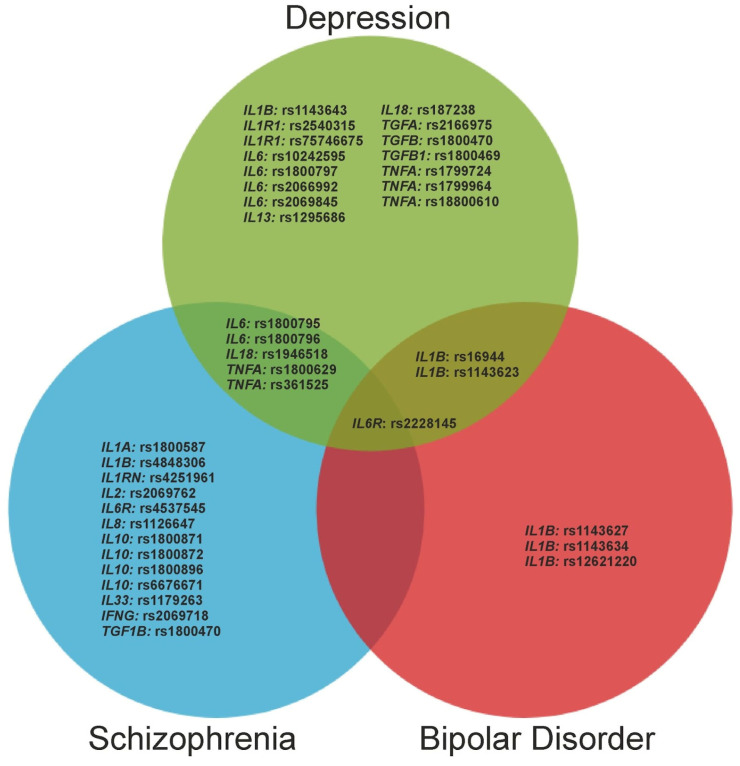
Venn diagram of summarized data on the association of single nucleotide polymorphisms of cytokine genes with depression, schizophrenia and bipolar disorder. Shared genetic associations are shown at the intersection of circles.

**Table 1 genes-14-01460-t001:** Association of cytokine and their receptor gene polymorphisms with depressive disorders.

Research	Analyzed Polymorphisms	Sample	Results
Kim et al. (2017) [29]	*TNFA:* rs1799724 (−850C/T; −308G/A); *IL1B*: rs16944 (−511C/T), +3953C/T	286 patients with PSD	−511C/T polymorphism was associated with primary depression and PSD at 2 weeks; higher TNF-α levels were associated with PSD at 2 weeks in in patients carried −850T allele.
Tartter et al. (2015) [30]	*IL6*: rs1800795 (−174G/C); *IL1B*: rs16944 (−511C/T); *TNF*: rs1800629 (−308G/A)	444 young adults whose exposure to chronic stress in the past 6 months; Australian cohort	Patients with the −174G allele had fewer depressive symptoms after interpersonal stress compared with CC homozygotes with equal exposure to interpersonal stress. The −511C allele in *IL1B* was associated with more severe depression after chronic interpersonal stress compared with TT homozygotes.
Lezheiko et al. (2018) [31]	*IL1B*: rs16944 (−511T/C); *TNFA*: rs1800629 (−308A/G)	139 patients with depression vs. 530 HS; Russian cohort	The −511T/C and −308A/G polymorphisms were associated with depression; CC genotype and GG genotype are the risk factors of depression.
McQuaid et al. (2019) [32]	*IL1B*: rs16944; *IL6*: rs1800795; *TNFA*: rs1800629	475 university students	Depressive symptoms were higher among individuals who experienced childhood adversity with the GG genotype of the *IL1B* rs16944.
Kovacs et al. (2016) [33]	*IL1B*: rs16944, rs1143643	1053 persons; Hungarian cohort	The rs16944*A allele was associated with childhood adversity increasing anxiety and depressive symptoms. The A allele of rs1143643 demonstrated protective effect against depressive symptoms after recent life stress.
Bialek et al. (2020) [18]	*IL1B*: rs1143623 (−1560G/C), rs1143627 (−118C/T); *IL1A:* rs17561 (c.340G/T); *TNFA:* rs1799964 (−1211T/C), rs1800629 (−488G/A)	270 patients with depression vs. 231 HS; Polish cohort	It was shown an association between the T allele and the TT genotype of rs1799964 *TNFA* and low effectiveness of pharmacotherapy; the C allele and CT genotype were associated with good response to therapy. Carryer of GC and CC genotypes of rs1143623 *IL1B* showed varying levels of disease severity ccording to the HDRS. The combined genotypes of rs1143627–rs17561, rs1143627–rs1799964 and rs1143623–rs1799964, decreased the risk of depression occurrence, rs1143627–rs1800629 increased the risk.
Kang et al. (2017) [34]	*IL1B: rs16944 (*−511C/T), +3953C/T	969 patients at 2 weeks after ACS, 711—at 1 year later	Depression during the acute ACS was associated with the −511T allele and the IL-1β levels. There was no association with depression in chronic ACS. There was no association with depression in the acute or chronic phase and the +3953C/T genotype.
Draganov et al. (2019) [35]	41 SNPs in *IL1B*, *IL2*, *IL6*, *IL6R*, *IL10*, *IL18*, *TNFA*, *IFNG*	153 patients with MDD	Polymorphic variant rs1143643 of *IL1B* was associated with MSM scores. Allelic distribution of rs57569414 *IL6R* demonstrates a trend to significance with MSM scores. Combinations of alleles of *IL1B* and *IL10* were associated with response to treatment.
Kim et al. (2013) [36]	*TNFA*: rs1800629 (−308G/A); *IL10:* rs1800896 (−1082A/G); *IFNG:* rs2430561 (+874T/A)	301 patients with MDD (204 attempted suicide, 97 not attempted suicide); Korean cohort	Among patients with MDD the TNFA −308GG genotype was associated with an increased risk of suicide; IL10-1082A/G were not associated with that risk.
Tsai et al. (2023) [37]	GWAS involving 684,616 SNPs	65 patients with TRD; Chinese cohort	Two SNPs (rs2540315 and rs75746675) in *IL1R1* were associated with a rapid (within 240 min) antidepressant effect of ketamine infusion in patients with TRD.
Kim et al. (2013) [38]	*TNFA*: rs1799724 (−850C/T); *IL1B*: rs16944 (−511C/T), +3953C/T; *IL6*: rs1800795 (−174G/C); *IL8*: −251T/A; *IL4:* +33T/C; *IL10:* rs1800896 (−1082A/G)	309 women with breast cancer at one week after surgery, 244 (79%)—at one year later.	*IL1B*-511TT was associated with depression at one week after surgery with breast cancer and persistent depression at one year follow-up.
Luckhoff et al. (2016) [39]	*TNFA:* rs1800629 *(*−308G/A)	94 patients with MDD vs. 97 HS; South African cohort	The rs1800629*A-allele in *TNFA* was associated with early-onset of MDD.
Lu et al. (2023) [40]	*IL6:* rs1800795; rs1800796	114 patients with depression vs. 110 HS; Han Chinese cohort	The CC genotype and the C allele of rs1800796 were associated with depression.
Golimbet et al. (2017) [41]	*IL4*: −589C/T; *IL6*: rs1800795 (−174G/C); *TNFA*: rs1800629 (−308G/A); *CRP*: −717A/G	78 male CHD patients with depression; 91—without depression; 127 HS; Russian cohort	The *IL6*-174G/C was associated with depression comorbid to CHD. The *IL4*-589C/T was associated with CHD. No association between the *TNFA*-308G/A and the *CRP*-717A/G with depression in CHD.
Kovacs et al. (2016) [42]	*IL6*: rs1800795	1053 volunteers; Hungarian cohort	The *IL6* rs1800795 in common with various stressors increases the risk of depression and has a greater impact measured by the ZSDS symptoms.
Gal et al. (2023) [43]	*IL6*: rs1800795	UK Biobank, n = 277 501	The rs1800795 was associated with recent stress on current depressive symptoms and lifetime depression.
Udina et al. (2013) [44]	*IL6*: rs1800795	385 patients with chronic hepatitis; Caucasian cohort	The rs1800795 *IL6* increases the risk of induced by IFN depression and anxiety. It was associated with fatigue rates in patients with chronic hepatitis C before treatment.
Zhang et al. (2016) [45]	*IL6*: 1800797	772 patients with MDD vs. 759 HS; Han Chinese cohort	Association between rs1800797 and the risk of MDD.
Maciukiewicz et al. (2015) [46]	Twenty SNPs in *IL1B*, *IL2*, *IL6*, *TSPO* and *BDNF*	MDD patients treated with duloxetine (n = 215) or placebo (n = 235)	Association *IL6* (−63G/A, rs2066992; +1984T/G, rs10242595) with response to duloxetine therapy in MDD patients. *IL6* rs2066992 and rs10242595 were associated with duloxetine response. The rs2066992 was associated with placebo response.
Khandaker at al. (2018) [47]	*IL6R:* rs2228145 (Asp358Ala)	9912 unselected participants from the ALSPAC birth cohort	Asp358Ala was associated with a reduced risk of severe depression and/or psychosis. Asp358Ala was not associated with total depression score and with the risk factors related with inflammation, depression or psychosis.
Dunn et al. (2013) [48]	104 SNPs and haplotypes in 15 cytokine genes	167 oncology patients with prostate, breast, lung, or brain cancer and 85 of their FCs	Significant associations of cytokine gene variants with trajectories of depressive symptoms in cancer patients and their FC have been identified. Two of these associations were in genes with anti-inflammatory functions (*IL1R2*, *IL10*), and one was with a gene with proinflammatory functions (*TNFA*).
Doong et al. (2015) [49]	82 SNPs in 15 genes of cytokine	398 breast cancer patients	Significant associations between *IL6* rs2069845, *IL13* rs1295686, and *TNFA* rs18800610 with a symptom cluster of pain, sleep disturbance, fatigue and depression.
Santos et al. (2016) [50]	*IL18:* rs1946518 (−607A/C), rs187238 (−137C/G)	80 MDD patients; Portuguese cohort	*IL18*-607A/C and *IL18*-137C/G were associated with the effect of the AD therapy. Patients carrying CA or AA genotypes of -607A/C and patients carrying GC or CC genotypes of -137C/G were significantly more prone to relapse after therapy and present a significantly lower time to relapse.
Sandoval-Carrillo et al. (2018) [51]	*TNFA:* rs1799724 (−857C/T), rs1800629 (−308G/A), rs361525 (−238G/A)	153 pregnant women with depression vs. 177 HS	The −857CT genotype increased the risk for depression. The −238GA genotype reduced the risk. No association between the −308G/A polymorphism and depression risk. The C857-G308-A238 haplotype was associated with a decrease of depression risk.
Saad et al. (2014) [52]	82 SNPs in 15 cytokine genes	155 patients with resilient and 180 patients with subsyndromal depressive symptom classes	In patients with breast cancer variation in three cytokine genes *IFNGR1* rs937626, *IL6* rs2069840, *TNFA* rs1799964, predicted membership in the Subsyndromal versus the Resilient class as well as age and functional status.
Bialek et al. (2020) [53]	*TGFB1*: rs1800469 (g.41354391A/G); *IRF*: rs2070729 (g.132484229C/A); *PTGS2*: rs5275 (186643058A/G); *PTGS2*: rs4648308 (g.186640617C/T); *TGF-α*: rs2166975 (g.70677994G/A); *IKBKB*: rs5029748 (g.42140549G/T).	80 patients with depression vs. 180 HS	The AG genotype of rs2166975 *TGFA* was associated with an increased risk of depression, the GG genotype reduced the risk. The AG genotype and G allele of the rs2166975 *TGFA* was associated with increased risk of depression development in men. Genotype rs1800469*AA of *TGFB1* was associated with earlier age of onset of the disease, GG genotype increased severity of the depressive episode.
Mihailova et al. (2016) [54]	*TNFA*, *TGFB*, *IL10*, *IL6*, *IFNG*	80 patients with depression vs. 50 HS; Bulgarian cohort	The *TGFB* + 869TT genotype (rs1800470) prevailed in patients compared with HS. The TT-GC combined genotype (+869T/C, +915G/C) was associated with disease recurrence.

Abbreviations: PSD: post-stroke depression; HS: healthy subjects; HDRS: Hamilton Depression Rating Scale; ACS: acute coronary syndrome; MDD: major depressive disorder; MSM: Mausdley Staging Method; SNP: single nucleotide polymorphism; GWAS: genome-wide association study; TRD: treatment-resistant depression; CHD: coronary heart disease; ZSDS: Zung Self-rating Depression Scale; FCs: Family Caregivers; AD: antidepressant therapy.

**Table 2 genes-14-01460-t002:** Association of cytokine and their receptor gene polymorphisms with schizophrenia.

Research	Analyzed Polymorphisms	Samples	Results
Srinivas et al. (2016) [67]	*IL1A*: rs1800587; *IL1B*: rs1143634, rs1143627, rs16944; *IL1RN*: rs2234663; *IL3*: rs31400, rs31480, rs40401; *IL4*: rs2243250, rs2070874; *IL6*: rs1800797, rs1800796, rs1800795; *IL10*: rs1800872, rs1800871, rs1800896; *IFNG*: rs2069718, rs2430561 *TNFA*: rs1800629, rs361525; *TGFB1*: rs1800471, rs1800470, rs1800469	248 SCZ vs. 24 HS; South Indian cohort	Only *IL1A* rs1800587, *IL6* rs1800796, *TNFA* rs361525, and *IFNG* rs2069718 polymorphisms were associated with SCZ. In silico analysis showed altered transcriptional activity for *IL1A* (rs1800587), *IL6* (rs1800796, rs1800795) and *TNFA* (rs361525).
Kapelski et al. (2016) [68]	*IL1A*: rs1800587, rs17561, rs11677416; *IL1B:* rs1143634, rs1143643, rs16944, rs4848306, rs1143623, rs1143633, rs1143627; *IL1RN*: rs419598, rs315952, rs9005, rs4251961	143 SCZ patients and their healthy parents	There is a trend toward an association of rs1143627, rs16944, rs1143623 in *IL1B* gene with the risk of SCZ. Alleles rs1143627*G, rs16944*A, and rs1143623*G were more frequently transmitted by parents to children with SCZ.
Paul-Samojedny et al. (2013) [69]	*IL6*: rs1800795; *TNFA*: rs1800629; *IL2*: rs2069762	115 SCZ vs. 135 HS; Polish cohort	Genotype TT and allele T of rs2069762 *IL2*, and genotype AA and allele A of rs1800629 *TNFA* were significantly associated with paranoid SCZ. Patients with haplotype CTA (rs1800795–rs1800629–rs2069762) showed higher scores on the Negative and General subscales of PANSS.
Kapelski et al. (2015) [70]	*IL1N*: rs1800587, rs17561; *IL1B*: rs1143634, rs1143643, rs16944, rs4848306, rs1143623, rs1143633, rs1143627; *IL1RN*: rs419598, rs315952, rs9005, rs4251961; *IL6*: rs1800795, rs1800797; *IL6R*: rs4537545, rs4845617, rs2228145; *IL10*: rs1800896, rs1800871, rs1800872, rs1800890, rs6676671; *IL10RA*: rs2229113, rs3135932; *TGFB1*: rs1800469, rs1800470	621 SCZ vs. 531 HS; Polish cohort	An association of genotype rs4848306*AG of *IL1B*, allele rs4251961*T of *IL1RN* gene, allele C and genotype AC of rs2228145 *IL6R*, allele T and genotyper CT of 4537545 *IL6R* with SCZ have been observed. Allele A of rs6676671 in *IL10* was associated with early age of onset.
Pu et al. (2021) [71]	*IL6*: rs1800795, rs1800796	113 SCZ vs. 110 HS; Han Chinese cohort	CC and CG genotypes of rs1800796 *IL6* was significantly associated with chronic SCZ
Frydecka et al. (2013) [72]	*IL2*: rs2069756 (−330T/G); *IL6*: rs1800795 (−174G/C); *IFNG*: rs2430561 (+874T/A); *TGF1B*: rs1800470 (+869T/C), rs1800471 (+916G/C)	151 SCZ vs. 279 HS; Caucasian cohort	The T carriers (CT and TT genotypes) of +869T/C were significantly more frequent in SCZ (especially in females) than in HS. Association of polymorphisms in *IL2*, *IL6* and *IFNG* genes with SCZ was not found.
Golimbet et al. (2022) [73]	*IL6*: rs1800795 (−174G/C); *IL10*: rs1800872 (−592C/A), rs1800896 (−1082G/A)	275 SCZ	Mean score on the AA subdomain of the Positive and Negative Syndromes Scale was higher in the AA genotype of *IL10*-592C/A compared with the CC genotype and in the GG genotype of *IL6*-174G/C compared with the AA genotype. AA score decreases with the number of the copies of an A allele of −1082G/A.
Ben Afia et al. (2020) [57]	*IL8*: rs4073, rs2227306, rs1126647	206 SCZ vs. 195 HS; Tunisian cohort	In the patients group it was observed an increased frequency of the T allele and the TT genotype of rs1126647. The T allele and the TT genotype of rs1126647 was associated with paranoid SCZ, and more specifically in females. The haplotypes TTT, ACT and TCT (rs4073-rs2227306-rs1126647) were associated with increased risk for paranoid SCZ, and only the TCT haplotype howed as a risk factor for SCZ.
Al-Asmary et al. (2014) [74]	*IL10*: rs1800896 (−1082A/G), rs1800871 (−819T/C), rs1800872 (−592A/C)	181 SCZ vs. 211 HS; Saudi Arabian cohort	Genotypes −1082GA, −819CC and −592CC are susceptible to SCZ, while genotypes −1082GG, −1082AA, −819CT and −592CA are resistant to SCZ.
Xiu et al. (2016) [75]	*IL10*: rs1800872 (−592A/C)	256 first-episode drug-naive SCZ vs. 540 HS; Han Chinese cohort	The A allele and AC genotype of −592A/C were associated with worse attentional performance in SCZ and reduced serum IL-10 levels. The AC genotype was associated with SCZ.
Subbanna et al. (2018) [76]	*IL17A*: rs2275913 (−197G/A)	221 SCZ vs. 223 HS	The AA genotype was associated with higher total scores of bizarre behavior and apathy in female SCZ patients. There was no significant difference in distribution of genotypes and alleles between SCZ and HS.
Zhang et al. (2016) [77]	*IL18*: rs1946518 (−607A/C)	772 SCZ vs. 775 HS; Han Chinese cohort	There were no significant differences in the distribution of the allele and genotype frequencies of −607A/C between SCZ and HS. The −607CC genotype was associated with higher PANSS general psychopathology subscore and the PANSS total score than both AC and AA genotypes.
Kordi-Tamandani et al. (2016) [78]	*IL33*: rs11792633	70 SCZ vs. 70 HS; Iranian cohort	CT and TT genotypes of rs11792633 significantly decreased the risk of SCZ.

Abbreviations: SCZ: schizophrenia patients; HS: healthy subjects; PANSS: the Positive and Negative Syndrome Scale; AA scores: avolition and apathy.

**Table 3 genes-14-01460-t003:** Association of cytokine and their receptor gene polymorphisms with bipolar disorder.

Research	Analyzed Polymorphisms	Samples	Results
Talaei et al. (2016) [55]	*IL1A*: rs1800587 (−889G/A); *IL1B*: rs1143634 (+3954C/T), rs16944 (−511C/T); *IL1RN*	48 BD vs. 47 HS; Iranian cohort	The frequency CC and CT genotype of *IL1B*-511C/T were significantly different between BD patients and healthy controls. The T allele of *IL1B*-511C/T was significantly more frequent in patients with a positive history of MDD. The T allele of *IL1B*+3954C/T was significantly more frequent in early onset BD patients.
Strenn et al. (2021) [79]	*IL1B*: rs1143623, rs1143627, rs16944, rs1143634	188 BD vs. 54 HS	The genotype distribution did not differ between patients with BD and the control group.
Pu et al. (2021) [71]	*IL1B*: rs1143643, rs1143634, rs1143627, rs16944, rs1143623, rs4848306, rs12621220	930 BD vs. 912 HS; Han Chinese cohort	The minor alleles of four polymorphisms of *IL1B* were associated with risk of BD (rs1143627*G, rs16944*A, rs1143623*G, rs12621220*T).
Shonibare et al. (2020) [80]	*IL1B*: rs16944	38 adolescents with BD vs. 32 HS	The rs16944 was associated with greater lateral occipital cortex surface area and volume in BD adolescents.
Sundaresh et al. (2018) [81]	*IL6*: rs1800795; *IL6R*: rs2228145	565 BD vs. 201 HS; French cohort	No association of the *IL6* rs1800795 and BD was found. The C allele and CC genotype of *IL6R* rs2228145 were associated with early onset of disease.

Abbreviations: BD: bipolar disorder; HS: healthy subjects; MDD—major depressive disorder.

## Data Availability

Not applicable.

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
