# Peer review of "Association of Single Nucleotide Polymorphisms of Cytokine Genes with Depression, Schizophrenia and Bipolar Disorder"

_genes, 2023, doi:10.3390/genes14071460_

Round 1
Reviewer 1 Report
The review “Association of single nucleotide polymorphisms of cytokine genes with depression, schizophrenia, and bipolar disorder” results interesting; however, needs some corrections.
This type of article should follow the PRISMA guide (http://www.prisma-statement.org/).
Add the flow diagram recommended by PRISMA and follow the checklist.
Materials and methods.
Please add the inclusion and exclusion criteria for the review and how studies were grouped for the syntheses.
Please, include filters and limits used
Describe the processes used to decide which studies were eligible.
Describe any methods required to prepare the data for presentation.
Discussion
Discuss the limitations of the evidence included in the review and the limitations of the review processes used.
Conclusion
The conclusion should be concise and clear; please rewrite
Reviewer 2 Report
The terminology can be used with the associated abbreviations (e.g. quality of life, schizophrenia = SCZ, post-traumatic stress disorder, and so on)
The Materials and Methods section may benefit from a better organization and structure, maybe creating subdivisions such as Database search, Inclusion and Exclusion criteria, Study selection, Number of entries, Number of results. This chapter can be expanded to a certain point.
On the other hand, the rest of the manuscript is relatively fairly written. Once the authors made the necessary revision, I think the paper can be accepted for publication.
Reviewer 3 Report
This is a very interesting review summarizing the current literature on the association of polymorphic variants of cytokine geners with the risk, severity and treatment response for severe mental disorders (e.g. schizophrenia, bipolar disorder, and major depression). The paper is well written and of interest for the journal; however, several changes are recommended.
ABSTRACT:
I recommend to add some words about the design of the review and the methods used. Is it a systematic or a narrative review? What search terms were used? The authors can included some words about inclusion and exclusion criteria in the abstract.
INTRODUCTION
1-Before summarizing the inflammatory hypothesis in schizophrenia, bipolar disorder and major depression, I recommend to introduce some aspects of the neurobiology of mental disorders, including biochemical findings, structural and functional brain findings, etc, as well as some words about neurotransmitter systems.
MATERIALS AND METHODS
1-This section should be expanded. I recommend to divide this section into at least three subsections. 1) Screening and selection processes. 2) Search strategy. 3) Assessment of studies.
RESULTS
Results have been presented according to each disorder. I recommend to add a brief paragraph explaining general results in the overall sample of studies.
CONCLUSIONS
The authors started this section with "This review synthetizes... This is not a good beginning for a conclusions section. The conclusions should be brief and concise and not a summary from the results.
Round 2
Reviewer 1 Report
The narrative review must meet quality control, please review SANRA (scale for the quality assessment of narrative review articles). https://researchintegrityjournal.biomedcentral.com/articles/10.1186/s41073-019-0064-8.
Apply SANRA criteria in this paper
Reviewer 2 Report
The overall quality and readability of your manuscript have been significantly improved. I have no more comments.
Author Response
Dear Reviewer,
The authors thank you for the positive evaluation of our manuscript.
Best regards